# Can we design the next generation of digital health communication programs by leveraging the power of artificial intelligence to segment target audiences, bolster impact and deliver differentiated services? A machine learning analysis of survey data from rural India

Jean Juste Harrisson Bashingwa [1] Diwakar Mohan [2] Sara Chamberlain [3] Kerry Scott [2] Osama Ummer [4] Anna Godfrey,[5] Nicola Mulder,[6] Deshendran Moodley,[7,8] Amnesty Elizabeth LeFevre [9]

For numbered affiliations see end of article.

**Correspondence to**
Dr Jean Juste Harrisson Bashingwa;
jeanjuste@aims.ac.za

## ABSTRACT

**Objectives** Direct to beneficiary (D2B) mobile health communication programmes have been used to provide reproductive, maternal, neonatal and child health information to women and their families in a number of countries globally. Programmes to date have provided the same content, at the same frequency, using the same channel to large beneficiary populations. This manuscript presents a proof of concept approach that uses machine learning to segment populations of women with access to phones and their husbands into distinct clusters to support differential digital programme design and delivery.

**Setting** Data used in this study were drawn from cross-sectional survey conducted in four districts of Madhya Pradesh, India.

**Participants** Study participant included pregnant women with access to a phone (n=5095) and their husbands (n=3842)

**Results** We used an iterative process involving K-Means clustering and Lasso regression to segment couples into three distinct clusters. Cluster 1 (n=1408) tended to be poorer, less educated men and women, with low levels of digital access and skills. Cluster 2 (n=666) had a mid-level of digital access and skills among men but not women. Cluster 3 (n=1410) had high digital access and skill among men and moderate access and skills among women. Exposure to the D2B programme 'Kilkari' showed the greatest difference in Cluster 2, including an 8% difference in use of reversible modern contraceptives, 7% in child immunisation at 10 weeks, 3% in child immunisation at 9 months and 4% in the timeliness of immunisation at 10 weeks and 9 months.

**Conclusions** Findings suggest that segmenting populations into distinct clusters for differentiated programme design and delivery may serve to improve reach and impact.

**Trial registration number** NCT03576157.

## STRENGTHS AND LIMITATIONS OF THIS STUDY

⇒ Segmenting populations into homogeneous groups can help to booster uptake of (direct to beneficiary) mobile health communication programmes.
⇒ The stepwise approach combining K-Means and Lasso regression is well superior compared with other approaches involving only either supervised or unsupervised machine learning to handle data from household surveys.
⇒ Our sample included men and women with a certain threshold of mobile phone access, possibly limiting the generalisability to populations with these characteristics.
⇒ Survey data included a large number of questions on mobile phone access and use, including observed digital skills, which to our knowledge are not widely available in India or elsewhere globally.
⇒ K-Means algorithm has certain limitations, including problems associated with random initialisation of the centroids which leads to unexpected convergence.

## INTRODUCTION

Digital health solutions have the potential to address critical gaps in information access and service delivery, which underpin high mortality.[1–9] Mobile health communication programmes, which provide information directly to beneficiaries, are among the few examples of digital health solutions to have scaled widely in a range of settings.[10 11] Historically, these solutions have been designed as 'blunt instruments'—providing the same content, with the same frequency, using the same digital channel to large target populations. While this approach has enabled

solutions to scale, it has contributed to variability in their reach and impact, due in part to differences in women's access to and use of mobile phones, particularly in low-income and middle-income countries.[12 13]

Despite near ubiquitous ownership of mobile phones at a household level, a growing body of evidence suggests that there is a substantial gap between men and women's ownership, access to and use of mobile phones.[14–16] In India, there is a 45% gap between women's reported access to a phone and ownership at a household level.[16] Variations in the size of the gap have been observed across states and urban/rural areas, and by sociodemographic characteristics, including education, caste and socioeconomic status.[16] Among women with reported access to a mobile phone, the gender gap further persists in the use of mobiles, in part because of patriarchal gender norms and limited digital skills.[17] Collectively, these gender gaps underscore the need to consider inequities in phone access and use patterns when designing and implementing direct to beneficiary (D2B) mobile health communication programmes.

Kilkari, designed and scaled by BBC Media Action in collaboration with the Ministry of Health and Family Welfare, is India's largest D2B mobile health information programme. When BBC Media Action transitioned Kilkari to the national government in April 2019, it had been implemented in 13 states and reached over 10 million women and their families.[3 18 19] Evidence on the programme's impact from a randomised control trial conducted in Madhya Pradesh, India, between 2018 and 2021, suggests that across study arms, Kilkari was associated with a 3.7% increase in modern reversible contraceptive use (RR: 1.12, 95% CI: 1.03 to 1.21, p=0.007), and a 2.0% decrease in the proportion of males or females sterilised since the birth of the child (RR: 0.85, 95% CI: 0.74 to 0.97, p=0.016).[3 19] The programme's impact on contraceptive use, however, varied across key population subgroups. Among women exposed to 50% or more of the Kilkari content as compared with those not exposed, differences in reversible method use were greatest for those in the poorest socioeconomic strata (15.8% higher), for those in disadvantaged castes (12.0% higher), and for those with any male child (9.9% higher).[3 19] Kilkari's overall and varied impact across beneficiary groups raises important questions about whether the differential targeting of women and their families might lead to efficiency gains and deepen impact.

In this manuscript, we argue that to maximise reach, exposure and deepen impact, the future design of mobile health communication solutions will need to consider the heterogeneity of beneficiaries, including within husband–wife couples, and move away from a one-size-fits all model towards differentiated programme design and delivery. Drawing from husbands' and wives' survey data captured as part of a randomised controlled trial (RCT) of Kilkari in Madhya Pradesh India, we used a three-step process involving K-Means clustering and Least Absolute Shrinkage and Selection Operator (Lasso) regression to segment couples into distinct clusters. We then assess differences in health behaviours across respondents in both study arms of the RCT. Findings are anticipated to inform future efforts to capture data and refine methods for segmenting beneficiary populations and in turn optimising the design and delivery of mobile health communication programmes in India and elsewhere globally.

## METHODS

### Kilkari program overview

Kilkari is an outbound service that makes weekly, stage-based, prerecorded calls about reproductive, maternal, neonatal and child health (RMNCH) directly to families' mobile phones, starting from the second trimester of pregnancy until the child is 1 year old. Kilkari is comprised of 90 min of RMNCH content sent via 72 once weekly voice calls (average call duration: 1 min, 15 s). Approximately 18% of cumulative call content is on family planning; 13% on child immunisation; 13% on nutrition; 12% on infant feeding; 10% on pregnancy care; 7% on entitlements; 7% on diarrhoea; 7% on postnatal care; and the remainder on a range of topics including intrapartum care, water and sanitation, and early childhood development. BBC Media Action designed and piloted Kilkari in the Indian state of Bihar in 2012–2013, and then redesigned and scaled it in collaboration with the Ministry of Health and Family Welfare between 2015 and 2019. Evidence on the evaluation design and programme impact are reported elsewhere.[20]

### Setting

Data used in this analysis were collected from four districts of the central Indian state of Madhya Pradesh as part of the impact evaluation of Kilkari described elsewhere.[3 19] Madhya Pradesh (population 75 million) is home to an estimated 20% of India's population and falls below national averages for most sociodemographic and health indicators.[21] Wide differences by gender and between urban and rural areas persist for wide range of indicators including literacy, phone access and health seeking behaviours. Among men and women 15–49 years of age, 59% of women (78% urban and 51% rural) were literate as compared with 82% of men in 2015–2016.[21] Among literate women, 23% had 10 or more years of schooling (44% urban and 14% rural).[21] Despite near universal access to phones at a household level, only 19% of women in rural areas and 50% in urban had access to a phone that they themselves could use in 2015.[21] Among pregnant women, over half (52%) of pregnant women received the recommended four antenatal care (ANC) visits in urban areas as compared with only 30% in rural areas.[21] Despite high rates of institutional delivery (94%) in urban areas, only 76% of women in rural areas reported delivering in a health facility in 2015.[21] These disparities underscore the population heterogeneity within and across Madhya Pradesh.

## Sample population

The samples for this study were obtained through cross-sectional surveys administered between 2018 and 2020 to women (n=5095) with access to a mobile phone and their husbands (n=3842) in four districts of Madhya Pradesh.[20] At the time of the first survey (2018–2019), the women were 4–7 months pregnant; the latter survey (2019–2020) reinterviewed the same women at 12 months post partum. Their husbands were only interviewed once, during the latter survey round. The surveys spanned 1.5 hours in length. In this analysis, modules on household assets and member characteristics; phone access and use, including observed digital skills (navigate interactive voice response (IVR) prompts, give a missed call, store contacts on a phone, open SMS, read SMS) were used to develop models. Data on practice for maternal and child health behaviours, including infant and young child feeding, family planning, pregnancy and post-partum care were used to explore the differential impact of Kilkari across clusters but not used in the development of clusters.[20]

### Approach to segmentation

Figure 1 presents a framework used for developing homogenous clusters of men and women in four districts of rural Madhya Pradesh India. Box 1 describes the steps undertaken at each point in the framework in detail. We started with data elements collected on phone access and use as well as population sociodemographic characteristics collected as part of a cross-sectional survey described elsewhere.[3 22] Unsupervised learning was undertaken using K-Means cluster and strong signals were identified. Strong signals were defined as variables that had at least a prevalence of 70% in one or more clusters and differed from another cluster by 50% or more. For example, 6% of men own a smart phone in Cluster 1, 88% in Cluster 2 and 75% in Cluster 3. Therefore, having a smart phone can be considered as a strong signal. Additional details

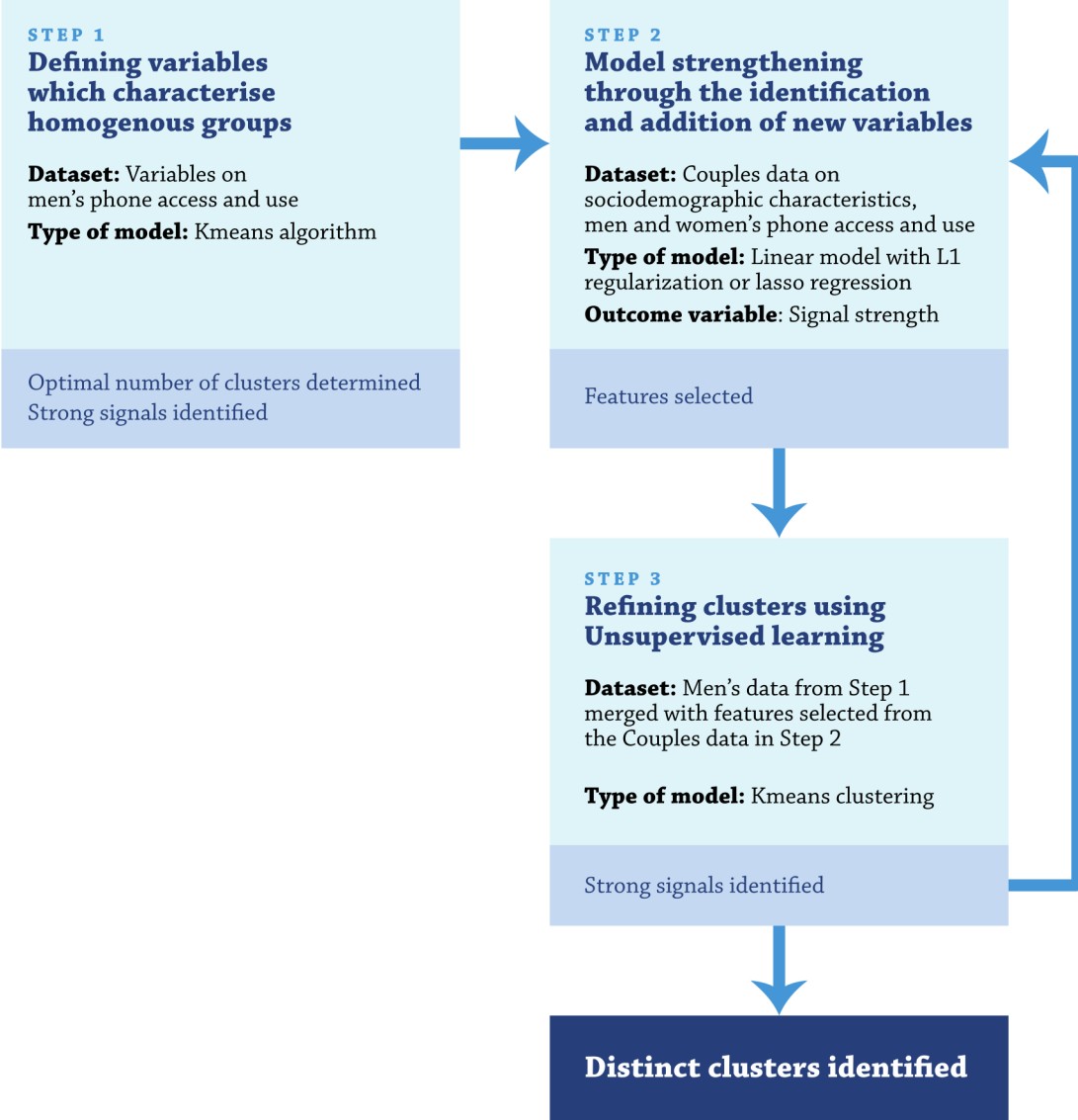

**Figure 1** Framework for segmentation analysis.

Box 1 Stepwise process for developing and refining a machine learning approach for population segmentation

Data collected from special surveys like the couple's dataset used here are relatively smaller in terms of sample size but large with regard to the number of data elements available. In such high-dimensional data, there are many irrelevant dimensions which can mask existing clusters in noisy data, making more difficult the development of effective clustering methods.[3 31] Several approaches have been proposed to address this problem. They can be grouped into two categories: static or adaptive *dimensionality reduction*, including principal components analysis[32 33] and *subspace clustering* consisting on selecting a small number of original dimensions (features) in some unsupervised way or using expert knowledge so that clusters become more obvious in the subspace.[34 35] In this study, we combined subspace clustering using expert knowledge and adaptive dimensionality reduction (online supplemental figure 1) to find subspace where clusters are most well separated and well defined. Therefore, as part of subspace clustering, we chose to start with couples' survey data, including variables related to sociodemographic characteristic, phone ownership, use and literacy (online supplemental table 1). Emergent clusters were overlapping. We decided to use men's survey data on phone access and use as a starting point.

**Step 1. Defining variables which characterise homogenous groups**

Analyses started with a predefined set of data elements captured as part of a men's cross-sectional survey including sociodemographic characteristics and phone access and use. K-Means clustering was used to identify clusters and the elbow method was used to define the optimal number of clusters. Strong signals were then identified. Variables which had at least a prevalence of 70% in one or more clusters and differed from another cluster by 50% or more were considered to have a strong signal.

**Step 2. Model strengthen through the identification and addition of new variables**

Once an initial model was developed drawing from the predefined set of data from the men's survey and strong signals were identified, we reviewed available data from the combined dataset (data from the men's survey and women's survey). Signal strength was used as an outcome variable or target in a linear regression with L1 regularisation or Lasso regression (Least Absolute Shrinkage and Selection Operator). Regularisation is a technique used in supervised learning to avoid overfitting. Lasso regression adds absolute value of magnitude of coefficient as penalty term to the loss function. The loss function becomes:

$$\text{Loss} = \text{Error}(y, y) + \alpha \sum_{i=1}^{N} |\omega_i|$$ where $\omega_i$ are coefficients of linear regression $y = \omega_1 x_1 + \omega_2 x_2 + \ldots + \omega_N x_N + b$.

Lasso regression works well for selecting features in very large datasets as it shrinks the less important features of coefficients to 0.[36 37] Merged women's survey and men's survey data were used as predictors for the regression, excluding variables related to heath knowledge and practices. We ended up with a sample of 3484 rows and 1725 variables after data preprocessing.

**Step 3. Refining clusters using supervised learning**

We then reran K-Means clustering with three clusters (K=3) using important features selected by Lasso regression. This methodology was used to refine the clusters and subsequently identify new strong signals. After step 3 was conducted, we repeated step 2, and kept on iteratively repeating step 2 and 3 until there was no gain in strong signals. Data preparation and results formatting have been conducted in R V.4.1.1,[38] K-Means clustering has been performed in Python V.3.8.5.[39]

are summarised in box 1. Once defined, we then explored differences in healthcare practices across study clusters among those exposed and not exposed to Kilkari within each cluster.

## Patient and public involvement

Patients were first engaged on identification in their households as part of a household listing carried out in mid/late 2018. Those meeting eligibility criteria were interviewed as part of the baseline survey, and ultimately randomised to the intervention and control arms. Prior to the administration of the baseline, a small number of patients were involved in the refinement of survey tools through qualitative interviews, including cognitive interviews, which were carried out to optimise survey questions, including the language and translation used. Finalised tools were administered to patients at baseline and endline, and for a subsample of the study population, additional interviews carried out over the phone and via qualitative interviews between the baseline and endline surveys. Unfortunately, because travel restrictions associated with COVID-19, findings were not disseminated back to community members.

## K-Means algorithm

As part of steps 1 and 3, K-Means algorithms were used (box 1). We chose to use K-Means algorithm because of its simplicity and speed to handle large dataset compared with hierarchical clustering.[23] A K-Means algorithm is one method of cluster analysis designed to uncover natural groupings within a heterogeneous population by minimising Euclidean distance between them.[24] When using a K-Means algorithm, the first step is to choose the number of clusters K that will be generated. The algorithm starts by selecting K points randomly as the initial centres (also known as cluster means or centroids) and then iteratively assigns each observation to the nearest centre. Next, the algorithm computes the new mean value (centroid) of each cluster's new set of observation. K-Means reiterates this process, assigning observations to the nearest centre. This process repeats until a new iteration no longer reassigns any observations to a new cluster (convergence). Four metrics have been used for the validation of clustering: within cluster sum of squares, silhouette index, Ray-Turi criterion and Calinski-Harabatz criterion. Elbow method was used to find the right K (number of clusters).[25] Figure 2 is a chart showing the within-cluster sum of squares (or inertia) by the number of groups (k value) chosen for several executions of the algorithm.

Inertia is a metric that shows how dissimilar the members of a group are. The less inertia there is, the more similarity there is within a cluster (compactness). The main purpose of clustering is not to find 100% compactness, it is rather to find a fair number of groups that could explain with satisfaction a considerable part of the data (k=3 in this case). Silhouette analysis helped to evaluate the goodness of clustering or clustering validation (figure 3). It can be used to study the separation

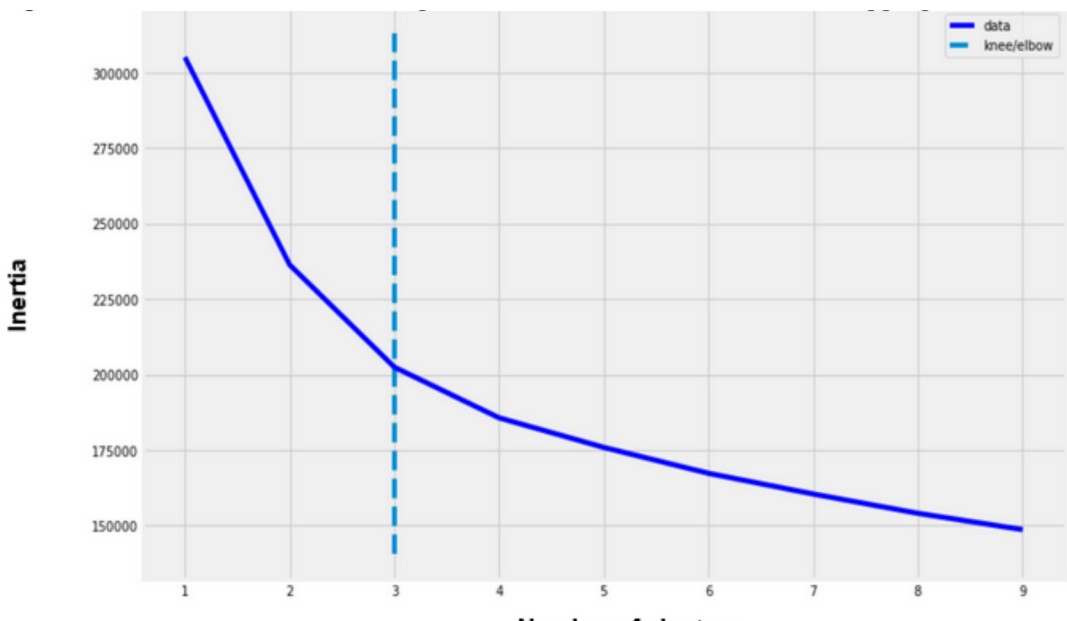

**Figure 2** Elbow method used to help decide ultimate number of clusters appropriate for the data.

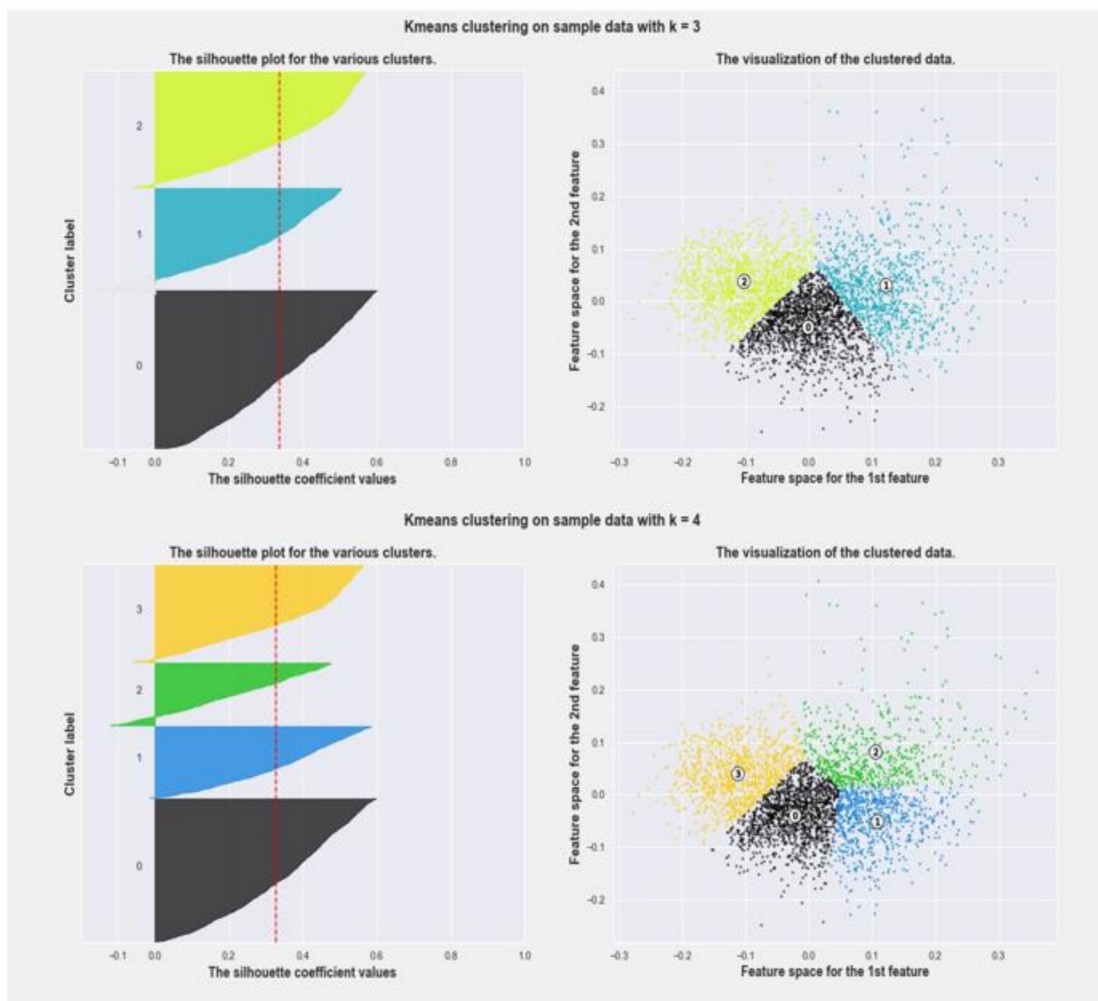

**Figure 3** Silhouette analysis for three and four clusters.

distance between the resulting clusters. The silhouette plot displays a measure of how close each point in one cluster is to points in the neighbouring clusters. This measure has a range of [−1, 1]. Silhouette coefficients near+1 indicate that the sample is far from the neighbouring clusters. A value of 0 indicates that the sample is very close to the decision boundary between two neighbouring clusters and negative values indicate that those samples might have been assigned to the wrong cluster. Figure 3 shows that choosing three clusters was more efficient than four for the data from the available surveys for two reasons: (1) there were less points with negative silhouettes and (2) the cluster size (thickness) was more uniform for three groupings. Other criteria used to evaluate quality of clustering are obtained by combining the 'within-cluster compactness index' and 'between-cluster spacing index'.[26] Calinski-Harabatz criterion is given by: $C(k) = \frac{\text{Trace}(B)\ (n-k)}{\text{Trace}(W)\ (k-1)}$ and Ray-Turi criterion is given by $r(k) = \frac{\text{distance}(W)}{\text{distance}(B)}$, where $B$ is the between-cluster covariance matrix (so high values of $B$ denote well-separated clusters) and $W$ is the within-cluster covariance matrix (so low values of $W$ correspond to compact clusters). They both ended up with same conclusions that three clusters were the best choice for the data we had. Online supplemental table 2 gives different metrics used and values obtained for various clusters.

## RESULTS

### Sample characteristics

Online supplemental tables 3A,B summarise the sample characteristics by cluster for men and women interviewed. Figure 4 and online supplemental table 4 presents select characteristics with 'strong signals' for each cluster.

Cluster 1 (n=1408) constitutes 40% of the sample population and was comprised of men and women with low levels of digital access and skills (figure 4). This cluster included the poorest segment of the sample population: 36% had a primary school or lower education and 40% were from a scheduled tribe/caste. Most men owned a feature (68%) or brick phone (22%); used the phone daily (89%); and while able to navigate IVR prompts (91%), only 29% were able to perform all of the five basic digital skills assessed. Women in this cluster similarly had lower levels of education as compared with other clusters (39% have primary school or less education); used feature (74%) or brick phones (8%); and had low digital skills (15% were able to perform the five basic digital skills assessed).

Cluster 2 (n=666; 19% of sample population) is comprised of men with mid-level and women with low digital access and skills. In this cluster, 75% of men owned smartphones, 65% were observed to successfully perform the five basic digital skills assessed and 36% could perform a basic internet search. Men in Cluster 2 also self-reported accessing videos from YouTube (84%)

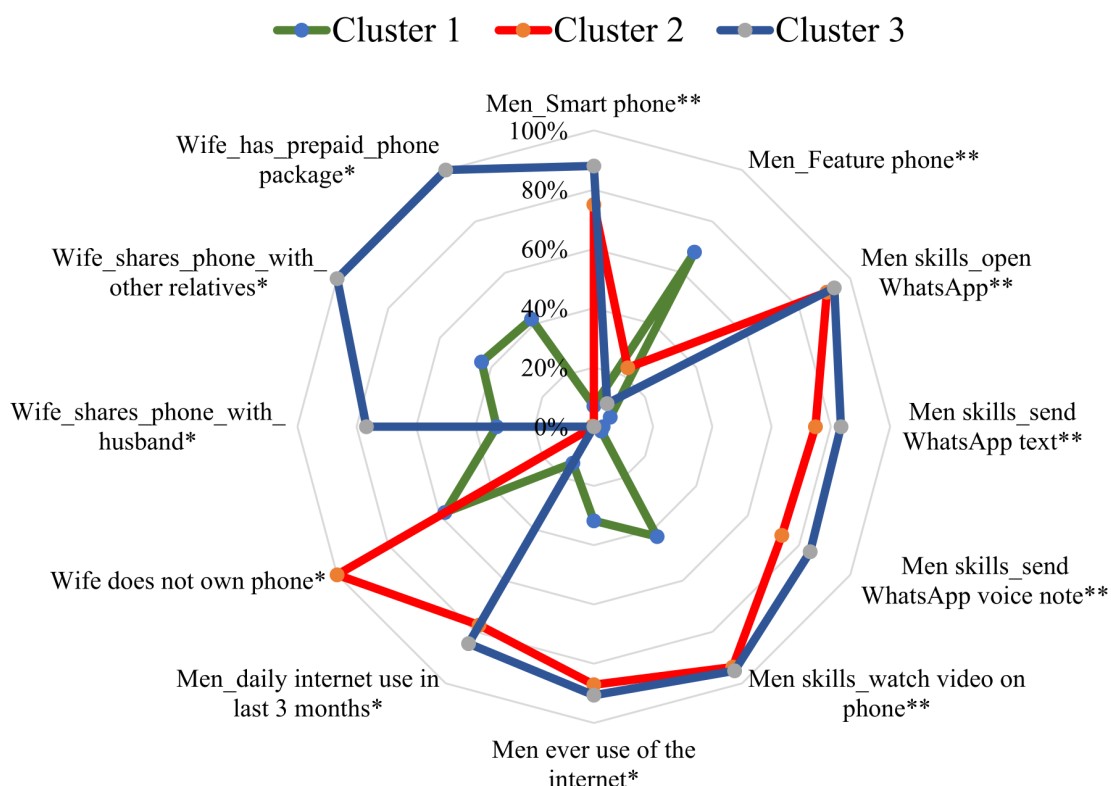

**Figure 4** Distribution of select characteristics with strong signals by Cluster. Variables which had at least a prevalence of 70% in one or more clusters and differed from another cluster by 50% or more were considered to have a strong signal (*reported by men interviewed; **observed by survey enumerators).

and using WhatsApp (95%). Women in Cluster 2 had low phone ownership; nearly half of women reported owning a phone (38% owned a phone and did not share it, 22% owned and shared a phone)—findings which contradict their husbands' reports of 0% women's phone ownership. Only 21% of women in this cluster were observed to be able to successfully perform the five basic digital skills assessed. However, based on husband's reporting of their wives' digital skills, 36% of women could search the internet, 37% used WhatsApp, and 66% watched shows on someone else's phone.

Cluster 3 (n=1410; 40% of sample population) is comprised of couples with high-level digital access among both husbands and wives, and lower-level digital skill among wives (figure 4). An estimated 67% of couples in this cluster were in the richer or richest socioeconomic strata, while 71% of men and 58% of women had high school or higher levels of education. Men in this cluster reported using the internet frequently (85%), were observed to own smart phones (88%) and had high levels of digital skills: 77% could perform the five basic digital skills assessed, 77% could perform a basic internet search and 85% could send a WhatsApp message. When reporting on their wife's digital access and skills, all men in this cluster reported that their wives' owned phones (100%), but often shared these phones with their husbands (77%), using them to watch shows (75%), search the internet (55%) or use WhatsApp (57%). However, a much lower level of women interviewed in this cluster were observed to own Feature (57%) or Smart phones (34%) and had moderate digital skills with 41% being able to successfully perform the five basic digital skills assessed.

### Differences in health outcomes by cluster

Table 1 presents differences in health outcomes by Cluster among those exposed and not exposed to Kilkari as part of the RCT in Madhya Pradesh. Findings suggest that the greatest impact was observed among those exposed to Kilkari in Cluster 2, which is the smallest cluster identified (19% of the sample population). Among this population, differences between exposed and not exposed were 8% for reversible modern contraceptive methods, 7% for immunisation at 10 weeks, 3% for immunisation at 9 months, and 4% for timely immunisation at 10 weeks and 9 months. Additionally, an 8% difference between exposed and not exposed was observed for the proportion of women who report being involved in the decision about what complementary foods to give child.

Among Clusters 1 and 3, improvements were observed among those exposed to Kilkari for a small number of outcomes. In Cluster 1, those exposed to Kilkari had a 3%–4% higher rate of immunisation at 6, 10, 14 weeks than those not exposed. In both Clusters 1 and 3, the timeliness of immunisation improved at 10 weeks among those exposed. No improvements were observed for use of modern reversible contraception in either cluster.

## DISCUSSION

Evidence on the impact of D2B mobile health communication programmes is limited but broadly suggests that they can cost-effectively improve some reproductive, maternal and child health practices. This analysis aims to serve as a proof of concept for segmenting beneficiary populations to support the design of more targeted mobile health communication programmes. We used a three-step iterative process involving a combination of supervised and unsupervised learning (K-Means clustering and Lasso regression) to segment couples into distinct clusters. Three identifiable groups emerge each with differing health behaviours. Findings suggest that exposure the D2B programme Kilkari may have a differential impact among the clusters.

### Implications for designing future digital solutions

Findings demonstrate that the impact of the D2B solution Kilkari varied across homogenous clusters of women with access to mobile phones and their husbands in Madhya Pradesh. Across delivery channels, our analysis indicates that mobile health communication could not be effectively delivered to husbands and wives in Cluster 1 using WhatsApp, because smartphone ownership and WhatsApp use in this cluster are negligible. IVR, on the other hand, could be used to reach couples in Cluster 1, but reach is likely to be sporadic because of high levels of phone sharing with others (78% among men and 57% among women). On the other hand, WhatsApp and YouTube are likely to be effective digital channels for communicating with both husbands and wives in Cluster 3, where most men and women own or use smartphones and WhatsApp.

Beyond delivery channels, study findings raise a number of important learnings for content development as well as optimising beneficiary reach and exposure. The creative approach to content created for Cluster 3, where 40% of women are from the richest socioeconomic status and only 17% have never been to school or have a primary school education or less, would need to be very different from the creative approach to content created for Cluster 1, where 53% have a poorest or poorer socioeconomic status, and 39% have never been to school or have a primary school education or less. Similarly, this analysis adds to qualitative findings[17] and provides important insights into how gender norms related to women's use of mobile phones may effect reach and impact. While few (13–15%) husbands indicated that 'adults' need oversight to use mobile phones, men's perceptions varied when asked about specific use cases. Across all Clusters, nearly half of husbands indicated that their wives needed permission to pick up phone calls from unknown numbers—an important insight for IVR programmes which may make outbound calls without prewarning to beneficiaries. In Clusters 1 and 2, 25% and 29% of husband's, respectively, report that their wives need permission to answer calls from health workers—as compared with 15% in Cluster 3. While restrictions on SMS and WhatsApp were lower

**Table 1** Differential impact of Kilkari exposure on family planning, infant feeding and immunisations per cluster

| | Cluster 1 | | | | | | Cluster 2 | | | | | | Cluster 3 | | | | | |
| | Not exposed | | | Exposed | | | Not exposed | | | Exposed | | | Not exposed | | | Exposed | | |
| | % | N | SE | % | N | SE | % | N | SE | % | N | SE | % | N | SE | % | N | SE |
| **Family planning** | | | | | | | | | | | | | | | | | | |
| Current modern family planning use | 42 | 269 | 0.02 | 41 | 316 | 0.018 | 42 | 130 | 0.028 | 44 | 157 | 0.026 | 50 | 340 | 0.019 | 51 | 368 | 0.019 |
| Reversible methods | 29 | 183 | 0.018 | 30 | 232 | 0.017 | 30 | 94 | 0.026 | 38 | 133 | 0.026 | 41 | 280 | 0.019 | 44 | 319 | 0.018 |
| Sterilised | 12 | 77 | 0.013 | 10 | 80 | 0.011 | 11 | 33 | 0.017 | 8 | 30 | 0.015 | 10 | 66 | 0.011 | 7 | 54 | 0.01 |
| Sterilised | 18 | 114 | 0.015 | 16 | 121 | 0.013 | 15 | 47 | 0.02 | 12 | 44 | 0.018 | 14 | 99 | 0.013 | 12 | 84 | 0.012 |
| **Infant and young child feeding** | | | | | | | | | | | | | | | | | | |
| Immediate breast feeding | 96 | 610 | 0.008 | 95 | 736 | 0.008 | 93 | 291 | 0.014 | 95 | 336 | 0.012 | 94 | 645 | 0.009 | 93 | 675 | 0.009 |
| Gave child semi solid food yesterday | 98 | 624 | 0.005 | 99 | 762 | 0.004 | 99 | 309 | 0.006 | 99 | 350 | 0.006 | 99 | 676 | 0.004 | 98 | 715 | 0.005 |
| Exclusive breast feeding | 6 | 39 | 0.01 | 6 | 48 | 0.009 | 7 | 21 | 0.014 | 8 | 28 | 0.014 | 6 | 43 | 0.009 | 7 | 51 | 0.009 |
| Fed child solid, semi-solid or soft foods the minimum number of times during the previous day | 54 | 344 | 0.02 | 55 | 423 | 0.018 | 62 | 193 | 0.028 | 64 | 228 | 0.025 | 66 | 450 | 0.018 | 65 | 469 | 0.018 |
| Minimum acceptable diet | 27 | 171 | 0.018 | 28 | 219 | 0.016 | 29 | 91 | 0.026 | 26 | 92 | 0.023 | 25 | 170 | 0.017 | 27 | 198 | 0.017 |
| Women involved in the decision about what complementary foods to give child | 89 | 569 | 0.012 | 92 | 708 | 0.01 | 82 | 256 | 0.022 | 90 | 319 | 0.016 | 88 | 604 | 0.012 | 87 | 634 | 0.012 |
| **Immunisation** | | | | | | | | | | | | | | | | | | |
| Fully immunised | 44 | 280 | 0.02 | 44 | 340 | 0.018 | 45 | 139 | 0.028 | 49 | 173 | 0.027 | 51 | 350 | 0.019 | 48 | 352 | 0.019 |
| Birth | 70 | 444 | 0.018 | 70 | 542 | 0.016 | 71 | 223 | 0.026 | 73 | 259 | 0.024 | 72 | 493 | 0.017 | 74 | 534 | 0.016 |
| 6 weeks | 75 | 475 | 0.017 | 78 | 600 | 0.015 | 78 | 242 | 0.024 | 79 | 280 | 0.022 | 77 | 528 | 0.016 | 78 | 568 | 0.015 |
| 10 weeks | 72 | 460 | 0.018 | 76 | 584 | 0.015 | 72 | 225 | 0.025 | 79 | 279 | 0.022 | 75 | 514 | 0.017 | 76 | 554 | 0.016 |
| 14 weeks | 68 | 432 | 0.019 | 71 | 550 | 0.016 | 74 | 230 | 0.025 | 74 | 263 | 0.023 | 75 | 511 | 0.017 | 75 | 541 | 0.016 |
| 9 months | 68 | 433 | 0.018 | 68 | 522 | 0.017 | 69 | 214 | 0.026 | 72 | 255 | 0.024 | 75 | 510 | 0.017 | 74 | 538 | 0.016 |
| Timeliness: birth | 69 | 438 | 0.018 | 67 | 515 | 0.017 | 68 | 213 | 0.026 | 69 | 246 | 0.025 | 70 | 477 | 0.018 | 72 | 525 | 0.017 |
| Timeliness: 6 weeks | 45 | 287 | 0.02 | 46 | 353 | 0.018 | 45 | 139 | 0.028 | 44 | 155 | 0.026 | 51 | 349 | 0.019 | 51 | 371 | 0.019 |
| Timeliness: 10 weeks | 25 | 162 | 0.017 | 28 | 217 | 0.016 | 23 | 71 | 0.024 | 27 | 94 | 0.024 | 31 | 213 | 0.018 | 34 | 248 | 0.018 |
| Timeliness: 14 weeks | 13 | 85 | 0.014 | 13 | 102 | 0.012 | 14 | 43 | 0.02 | 14 | 51 | 0.019 | 19 | 131 | 0.015 | 22 | 162 | 0.015 |
| Timeliness: 9 months | 14 | 89 | 0.014 | 13 | 99 | 0.012 | 12 | 37 | 0.018 | 16 | 55 | 0.019 | 18 | 126 | 0.015 | 17 | 126 | 0.014 |

than making or receiving calls, these channels are less viable given women's limited access to smartphones, low literacy and digital skills. Overall, men's perceptions on the restrictions needed on the receipt and placement of calls by women was lower for Cluster 3. However, despite the relative wealth of beneficiaries in Cluster 3 (67% were in the richer or richest socioeconomic strata), 48% of women had zero balance on their mobile phones at the time of interview. Collectively, these findings highlight the immense challenges which underpin efforts to facilitate women's phone access and use. They too underline the criticality of designing mobile health communication content for couples, rather than just wives to ensure the buy-in of male gatekeepers, and for continuing to prioritise face to face communication with women on critical health issues.

## Approach to segmentation

Data in our sample were captured as part of special surveys carried out through the impact evaluation of Kilkari. Future programmes may be tempted to apply the approach undertaken here to existing datasets, including routine health information systems or other forms of government tracking data. In the India context, while these data are likely to be less costly than special surveys, they are comparatively limited in terms of data elements captured—particularly in terms of data ownership of different types of mobile devices, digital skill levels and usage of specific applications or social media platforms. Data quality may also be a significant issue in existing datasets . For example, we estimate that SIM change in our study population was 44% over a 12-month period—a factor which when coupled with the absence of systems to update government tracking registries raises important questions about who is retained in these databases, and therefore able to receive mobile health communications—and who is missing. Among the variables used, men's phone access and use were most integral to developing distinct clusters. We recommend that future surveys seeking to generate data for designing digital services for women ensure that data elements are captured on men's phone access and use practices as well as their perception of their wife's phone access and use.

In addition to underlying data, our analytic approach differed from other segmentation analyses. Our work is relatively new in global health literature related to digital health programmes that are positioned as D2B programmes. While similar ML models are being tested in various domains related to public health, they consist exclusively of unsupervised learning[27 28] or supervised learning,[1 6 29 30] this analysis is the first of its kind focusing on the use of a combination of supervised and unsupervised learning to identify homogenous clusters for targeting of digital health programmes. Data collected from special surveys like the couple's dataset used here are comparatively smaller in terms of sample size but large with regard to the number of data elements available. An alternative approach to that described in this manuscript might be to develop strata based on population characteristics. Indeed, findings from the impact evaluation published elsewhere suggest that women with access to phones in the most disadvantaged sociodemographic strata (poorest (15.8% higher) and disadvantaged castes (12% higher)) had greater impact when exposed to 50% or more of the Kilkari content as compared with those not exposed. With an approach to segmentation based on these strata of highest impact, we know and understand what divides or groups respondents (eg, socioeconomic status, education) but this may not be enough when they do not explain the underlying reasons for change. In the approach used here, the study population is segmented using multiple characteristics (sociodemographic, digital access and use) simultaneously. The results are clusters comprised of individuals with mixed sociodemographic characteristics which may help to explain the reduced impact observed on health outcomes. Designing a strategy based on previously known/identifiable strata alone has been the basis of targeting in public health but has not maximised reach, exposure and effect to its fullest potential. The approach used here may better group beneficiaries based on their digital access and use characteristics which may serve to increase reach and exposure. However, further research is needed to determine how to deepen impact within these digital clusters.

## CONCLUSIONS

Study findings sought to identify distinct clusters of husbands and wives based on their sociodemographic, phone access and use characteristics, and to explore the differential impact of a maternal mobile messaging programme across these clusters. Three identifiable groups emerge each with differing levels of digital access and use. Descriptive analyses suggest that improvements in some health behaviours were observed for a greater number of outcomes in Cluster 2, than in Clusters 1 and 3. These findings suggest that one size fits all mobile health communications solutions may only engage one segment of a target beneficiary population, and offer much promise for future D2B and other digital health programmes which could see greater reach, exposure and impact through differentiated design and implementation. More quantitative and qualitative work is needed to better understand factors driving the differences in impact and what is likely to motivate adoption of target behaviours in different clusters. Our work opens up a new avenue of research into better targeting of beneficiaries using data on variety of domains including sociodemographics, mobile phone access and use. Future work will entail evaluation of the actual platform used for targeting and delivery of the programme in pilot projects. Successful pilots can be scaled up to larger swathes of the population in India and similar setting around the world.

**Author affiliations**
[1]School of Public Health, University of the Witwatersrand, Johannesburg, South Africa
[2]Department of International Health, Johns Hopkins University Bloomberg School of Public Health, Baltimore, Maryland, USA
[3]Independent Consultant, Digital Health & Gender, Delhi, India
[4]Oxford Policy Management, New Delhi, India
[5]BBC Media Action, London, UK
[6]Computational Biology Division, Department of Integrative Biomedical Sciences, IDM, University of Cape Town Faculty of Heath Sciences, Cape Town, South Africa
[7]Department of Computer Science, University of Cape Town, Cape Town, South Africa
[8]Centre for Artificial Intelligence Research, University of Cape Town, Cape Town, South Africa
[9]Division of Public Health Medicine, University of Cape Town, School of Public Health, Cape Town, South Africa

**Acknowledgements** We thank the women and families of Madhya Pradesh who generously gave of their time to support this work. We are humbled by the opportunity to convey their perspectives and experiences. We additionally are grateful to Dr Rajani Ved at the National Health Systems Resource Centre for her support. This work was made possible by the Bill and Melinda Gates Foundation. We thank Diva Dhar, Suhel Bidani, Rahul Mullick, Dr Suneeta Krishnan, Dr Neeta Goel and Dr Priya Nanda for believing in us and giving us this opportunity. We additionally wish to thank BBC Media Action teams in India and London for their partnership, collaboration, and inputs, including Payal Rajpal and Varinder Gambhir. We too are grateful to the larger team of enumerators from OPM-India who worked tirelessly over many months to implement the surveys that form the backbone of our analyses. We additionally thank Prabal Singh, Vinit Pattnaik at OPM and Alain Labrique, Smisha Agarwal and Erica Crawford at Johns Hopkins University for their support. Lastly, our figures have been beautified by Dan Harder of the Creativity Club UK. We thank him for his work.

**Contributors** JJHB conducted the analysis and wrote the paper with AEL and inputs from DM, SC, and other authors. AEL is the guarantor of the content, the overall study PI, helped to secure the funding, led the design of the study tools, supported oversight of field work and analysis and wrote the manuscript with JJHB and DM. AEL is responsible for the overall content as guarantor. DM helped to secure funding, helmed the study design including sampling and randomisation, helped draft study tools, provided input to data analysis and edited the manuscript. SC helped to secure the funding, draft and review study tools, interpret data analyses and study findings and edit the manuscript. AG, KS, helped to draft and review study tools, interpret data analyses and study findings and edit the manuscript. OU help to revise study tools, interpret data analyses and edited the manuscript. NM is the UCT study PI and provided input to study design, oversight to the analysis and interpretation and edited the manuscript.

**Funding** Bill and Melinda Gates Foundation grant number OPP1179252.

**Competing interests** All authors have completed the Unified Competing Interest form (available on request from the corresponding author) and declare that the research reported was funded by the Bill and Melinda Gates Foundation. AG and SC are employed by BBC Media Action; one of the entities supporting program implementation. The authors do not have other relationships and are not engaged in activities that could appear to have influenced the submitted work.

**Patient and public involvement** Patients and/or the public were not involved in the design, or conduct, or reporting or dissemination plans of this research.

**Patient consent for publication** Consent obtained directly from patient(s).

**Ethics approval** Institutional Review Boards from the Johns Hopkins Bloomberg School of Public Health in Baltimore, Maryland USA and Sigma Research and Consulting in Delhi, India provided ethical clearance for study activities. Verbal informed consent was obtained from all study participants.

**Provenance and peer review** Not commissioned; externally peer reviewed.

**Data availability statement** Data are available upon reasonable request to the Study PI, Amnesty LeFevre (aelefevre@gmail.com).

**ORCID iDs**
Jean Juste Harrisson Bashingwa http://orcid.org/0000-0003-1433-4397
Diwakar Mohan http://orcid.org/0000-0002-7532-366X
Sara Chamberlain http://orcid.org/0000-0003-4785-6482
Kerry Scott http://orcid.org/0000-0003-3597-9637
Osama Ummer http://orcid.org/0000-0002-4189-5328
Amnesty Elizabeth LeFevre http://orcid.org/0000-0001-8437-7240

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
