## [Reviewer comments · BMJ Open]

ARTICLE DETAILS

TITLE (PROVISIONAL)	Can we design the next generation of digital health communication programs by leveraging the power of artificial intelligence to segment target audiences, bolster impact, and deliver differentiated services? A machine learning analysis of survey data from rural India
AUTHORS	Bashingwa, Jean; Mohan, Diwakar; Chamberlain, Sara; Scott, Kerry; Ummer, Osama; Godfrey, Anna; Mulder, Nicola; Moodley, Dshen; LeFevre, Amnesty

VERSION 1 – REVIEW

REVIEWER	Maniruzzaman, Md. Khulna University, Statistics Discipline
REVIEW RETURNED	02-Jul-2022

GENERAL COMMENTS	Review Report Manuscript ID: BMJ open-2022-063354 Thank you so much for allowing me to review the manuscript “Can we design the next generation of digital health communication programs by leveraging the power of artificial intelligence to segment target audiences, bolster impact, and deliver differentiated services? A machine learning analysis of survey data from rural India”. The manuscript is interesting, well-written, well-organized, and technically sounds. I read the manuscript thoroughly and my specific comments are as follows: 1. The methodology part of ridge regression is missing.2. Why the authors choose ridge regression? Is the database have multi-collinearity problems? It needs to explain clearly.3. What are strong points of this study and added its before limitations?4. How the authors select the independent or predictor variables for diabetes? If they select these based on the previous literatures, please provided some references?5. I encourage the authors to make a section of “Related Work”.6. Many of the references appeared dated -over 10 years since publication - and I encourage you to consult more recent work in your review.
--

REVIEWER	Chow, James Princess Margaret Hospital Cancer Centre, Medical Physics
REVIEW RETURNED	29-Sep-2022

GENERAL COMMENTS	Referee Report Manuscript Number: bmjopen-2022-063354 Title: Can we design the next generation of digital health communication programs by leveraging the power of artificial intelligence to segment target audiences, bolster impact, and deliver differentiated services? A machine learning analysis of survey data from rural India submitted to BMJ Open by Bashingwa et al This work used machine learning (ML) to segment populations of women with access to phones and their husbands into various clusters to support the D2B mobile health communication programs in India. There were 5,095 pregnant women access to a phone with their husbands (n = 3,842). From the study, the authors concluded that segmenting populations using ML (K-mean algorithm) might improve the reach and impact for the differentiated program design and delivery. I have the following concerns in this work:  1. Title: The title of this manuscript may be too long and the authors may want to rephrase or shorten it. 2. Abstract: Sum of Cluster 1, 2 and 3 is equal to 3,484. This is less than the number of participant in the program. 3. Introduction: When mentioning recent advances of AI/ML on the health communication programs/health care, please consider using more updated references such as Xu et al (JIMR Cancer 2021;7:e27850) and Siddique et al (Encyclopedia 2021;1:220). 4. Methods, Kilhari program overview: It is good to provide a website link or reference for the Kilhari program. 5. Methods, Setting: It is good to provide a table for the background and details of the data sources for the four districts. 6. Methods, Approach to segmentation: There are many ML methods. Please justify why the authors selected the “K-means” clustering algorithm in this study. Moreover, please provide more information about the software and programming platform to run the ML algorithm. 7. Figure 1: For the equation: $Loss = Error(y,y) + \alpha \sum N_i = 1 \omega_i$, please explain why there are two same variables of “y” in Error ()? 8. Table 1: Please provide statistical analysis such as SD. 9. Discussion, Implications for designing future digital solutions: I think it is necessary to define mobile and smartphone here. It is because mobile phone may not be smartphone which participant can install Apps (e.g. WhatsApp) to the phone. 10. Discussion, Limitations: Same as 9, authors need to explain why the program can only work on mobile phone but not local/home phone without connecting to the network. 11. Conclusions, The authors may want to mention what will be the future work in this study, but not just summarized the findings.
--

VERSION 1 – AUTHOR RESPONSE

Reviewer 1	
Reviewer comment	Response to comment
Thank you so much for allowing me to review the manuscript “Can we design the next generation of digital health communication programs by leveraging the power of artificial intelligence to segment target audiences, bolster impact, and deliver differentiated services? A machine learning analysis of survey data from rural India”. The manuscript is interesting, well-written, well-organized, and technically sounds. I read the manuscript thoroughly and my specific comments are as follows:	Thank you for taking the time to review our paper.
1. The methodology part of ridge regression is missing.	Thank you for pointing this out. We realize that we have been using two denominations -- (Lasso and ridge regression). While the two use different regularization techniques, they operate quite similarly. We have corrected in the manuscript by keeping the denomination “Lasso regression” described in Box 1.
2. Why the authors choose ridge regression? Is the database have multi-collinearity problems? It needs to explain clearly.	In household surveys, they collect lot of information. The datasets are large, and collinearity is inevitable between certain variables. However, as explained previously, we used LASSO regression, which works well for features selection as described in Box1. Since the focus was on prediction rather than causal explanations, the multicollinearity becomes less of an issue as only the predictive power of the variables are considered. Variables that do not add to the predictive ability beyond that of other variables are excluded by the model.
3. What are strong points of this study and added its before limitations?	We have added a section Strengths and Limitation of this study after the abstract as suggested by the editor. We added the following text: “Strengths and limitations of this study: Strengths  • The step-wise approach combining K-means and Lasso regression is well superior compared to other approaches involving only either supervised or unsupervised machine learning to handle data from household surveys.

	 Findings suggest that segmenting populations into homogeneous groups can help to booster uptake of (D2B) mobile health communication programs. Limitations  The analysis included only those with a certain (higher than that of general population) level of access to mobile phones - survey respondents were required to have access to a mobile phone (own a phone or have a phone they can use). While populations without a high level of access to phones may have different findings, our analysis presents what is typical of populations that are enrolled in direct to beneficiary programs. K-means algorithm has certain limitations, including problems associated with random initialization of the centroids which leads to unexpected convergence. Also, the empirical nature of the methods may limit the generalisability of the exact variables to other settings. ”
4. How the authors select the independent or predictor variables for diabetes? If they select these based on the previous literatures, please provided some references?	Thank you for this comment. Diabetes is not a health condition considered in this analysis.
5. I encourage the authors to make a section of “Related Work”.	Our work is relatively new in global health literature related to digital health programs that are positioned as direct to beneficiary programs. While similar ML models are being tested in various domains such as pregnancy outcome prediction primarily with supervised learning methods, our analysis is the first of its kind focusing on the use of a combination of supervised and unsupervised learning to identify homogenous clusters for targeting of digital health programs. Raj A, Dehingia N, Singh A, McDougal L, McAuley J. Application of machine learning to understand child marriage in India. SSM Popul Health. 2020 Dec 5;12:100687. doi: 10.1016/j.ssmph.2020.100687. PMID: 33335970; PMCID: PMC7732880. Dey AK, Dehingia N, Bhan N, Thomas EE, McDougal L, Averbach S, McAuley J, Singh A, Raj A. Using machine learning to understand determinants of IUD use in India: Analyses of the National Family Health Surveys (NFHS-4). SSM Popul Health. 2022 Sep 29;19:101234. doi:

	10.1016/j.ssmph.2022.101234. PMID: 36203476; PMCID: PMC9529578. Das R, Saleh S, Nielsen I, Kaviraj A, Sharma P, Dey K, Saha S. Performance analysis of machine learning algorithms and screening formulae for β-thalassemia trait screening of Indian antenatal women. Int J Med Inform. 2022 Nov;167:104866. doi: 10.1016/j.ijmedinf.2022.104866. Epub 2022 Sep 16. PMID: 36174416.
6. Many of the references appeared dated -over 10 years since publication - and I encourage you to consult more recent work in your review.	Thank you for your comment. We have added more recent publications in the introduction.
Reviewer 2	
Reviewer comment	Response to comment
This work used machine learning (ML) to segment populations of women with access to phones and their husbands into various clusters to support the D2B mobile health communication programs in India. There were 5,095 pregnant women access to a phone with their husbands (n = 3,842). From the study, the authors concluded that segmenting populations using ML (K-mean algorithm) might improve the reach and impact for the differentiated program design and delivery. I have the following concerns in this work:	Thank you for your thoughtful review.
1. Title: The title of this manuscript may be too long and the authors may want to rephrase or shorten it.	Thank you. This concern is noted.
2. Abstract: Sum of Cluster 1, 2 and 3 is equal to 3,484. This is less than the number of participant in the program.	In the analysis we used couples' data. After merging women dataset and men dataset, we ended up with 3484. More description is given in Box1
3. Introduction: When mentioning recent advances of AI/ML on the health communication programs/health care, please consider using more updated references such as Xu et al (JIMR Cancer 2021;7:e27850) and Siddique et al (Encyclopedia 2021;1:220).	We have added more recent publication in the introduction.

4. Methods, Kilkari program overview: It is good to provide a website link or reference for the Kilkari program.	Thank you for your comment. We have bolstered the references to include additional publications describing the program. These include A.E. LeFevre, N. Shah, K. Scott, S. Chamberlain, O. Ummer, J.J.H. Bashingwa, A. Chakraborty, A. Godfrey, P. Dutt, R.J.B.g.h. Ved, The impact of a direct to beneficiary mobile communication program on reproductive and child health outcomes: a randomised controlled trial in India, 6(Suppl 5) (2022) e008838. LeFevre A, Agarwal S, Chamberlain S, Scott K, Godfrey A, Chandra R, Singh A, Shah N, Dhar D, Labrique A, Bhatnagar A, Mohan D. Are stage-based health information messages effective and good value for money in improving maternal newborn and child health outcomes in India? Protocol for an individually randomized controlled trial. Trials. 2019 May 15;20(1):272. doi: 10.1186/s13063-019-3369-5. PMID: 31092278; PMCID: PMC6521473.
5. Methods, Setting: It is good to provide a table for the background and details of the data sources for the four districts.	We are constrained by word limits and have provided variables (sample characteristics) included in the last model in Supplementary Tables 3a and 3b. Readers who have interest to know more on kilkari can use these references which have been added in the manuscript: A.E. LeFevre, N. Shah, K. Scott, S. Chamberlain, O. Ummer, J.J.H. Bashingwa, A. Chakraborty, A. Godfrey, P. Dutt, R.J.B.g.h. Ved, The impact of a direct to beneficiary mobile communication program on reproductive and child health outcomes: a randomised controlled trial in India, 6(Suppl 5) (2022) e008838. LeFevre A, Agarwal S, Chamberlain S, Scott K, Godfrey A, Chandra R, Singh A, Shah N, Dhar D, Labrique A, Bhatnagar A, Mohan D. Are stage-based health information messages effective and good value for money in improving maternal newborn and child health outcomes in India? Protocol for an individually randomized controlled trial. Trials. 2019 May 15;20(1):272. doi: 10.1186/s13063-019-3369-5. PMID: 31092278; PMCID: PMC6521473.
6. Methods, Approach to segmentation: There are many ML methods. Please justify why the authors selected the “K-means” clustering algorithm in this study. Moreover, please provide more information about the software and programming platform to run the ML algorithm.	Your point is very well taken. We have added the following sentences “We choose to use K-means algorithm because of its simplicity and speed to handle large dataset compared to hierarchical clustering” and

	“Data preparation and results formatting have been conducted in R 4.1.1, K-means clustering has been performed in python 3.8.5.”
7. Figure 1: For the equation: $Loss = Error(y, \hat{y}) + \alpha \sum N_i = 1 \omega_i$, please explain why there are two same variables of “y” in Error ()?	There are not the same variables (one has a hat). It must be an error produced by the system when it converts word to PDF document. y-- stands for the actual value and \hat{y} is the estimated value (or the predicted value in machine learning jargon)
8. Table 1: Please provide statistical analysis such as SD.	We have added the standard error (SE) on Table 1
9. Discussion, Implications for designing future digital solutions: I think it is necessary to define mobile and smartphone here. It is because mobile phone may not be smartphone which participant can install Apps (e.g. WhatsApp) to the phone.	Thank you. Participants in this study are either mobile phone owner or have access to it. In the supplementary table 3a and 3b, we described the type of phone owned by participants. The direct to beneficiary programs (eg Kilkari) rely on IVR messaging which work on basic phones with a network connection. Smartphone usage, while on the rise, is still quite low among the target population that are the focus of such D2B programs. In the short-term, digital solutions will need to cater to basic mobile phone users while laying the groundwork for future use of smartphones.
10. Discussion, Limitations: Same as 9, authors need to explain why the program can only work on mobile phone but not local/home phone without connecting to the network.	The design of the program is such that mobile phone numbers are captured and used for the Kilkari program. Local / home phones are almost non-existent among the populations beyond urban areas or businesses.
11. Conclusions, The authors may want to mention what will be the future work in this study, but not just summarized the findings.	Our work opens up a new avenue of research into better targeting of beneficiaries using data on variety of domains including socio-demographics, mobile phone access and use. Future work will entail evaluation of the actual platform used for targeting and delivery of the program in pilot projects. Successful pilots can be scaled up to larger swathes of the population in India and similar setting around the world.

VERSION 2 – REVIEW

REVIEWER	Maniruzzaman, Md. Khulna University, Statistics Discipline
REVIEW RETURNED	12-Nov-2022

GENERAL COMMENTS	The authors have answered the remarks.
--

REVIEWER	Chow, James Princess Margaret Hospital Cancer Centre, Medical Physics
REVIEW RETURNED	07-Nov-2022

GENERAL COMMENTS	I am satisfied with the corrections made by the authors as per my comments. I have no further question.
---

VERSION 2 – AUTHOR RESPONSE

Reviewer 1	
Reviewer comment	Response to comment
The authors have answered the remarks.	Thank you for taking the time to review our paper.
Reviewer 2	
Reviewer comment	Response to comment
I am satisfied with the corrections made by the authors as per my comments. I have no further question.	Thank you for your thoughtful review.